# "I can be a source of motivation": Perspectives from stakeholders of the I'mPossible fellowship, a peer-led differentiated service delivery model for adolescents with perinatally acquired HIV in India

Siddha Sannigrahi[1], Michael Babu Raj[2], Babu Seenappa[2], Ashley A. Sharma[3], Suhas Reddy[2], Esha Nobbay[4], Aastha Kant[5], Satish Kumar SK[6], Baldeep K. Dhaliwal[1], Lakshmi Ganapathi[7], Anita Shet[1]*

1 Department of International Health, Johns Hopkins Bloomberg School of Public Health, Baltimore, Maryland, United States of America, 2 Monitoring, Evaluation and Learning Division, RISHI Foundation, Bengaluru, India, 3 Robert Larner, M.D. College of Medicine, University of Vermont, Burlington, Vermont, United States of America, 4 Department of Psychiatry, Adichunchanagiri Institute of Medical Sciences, B G Nagara, India, 5 Johns Hopkins India Private Limited, New Delhi, India, 6 Y.R. Gaitonde Centre for AIDS Research and Education (YRGCARE), Chennai, India, 7 Division of Pediatric Infectious Diseases and Global Health, Massachusetts General Hospital, Harvard Medical School, Boston, Massachusetts, United States of America

* ashet1@jhu.edu

## Abstract

Youth living with perinatally acquired HIV (APHIV) in India face layered challenges; peer-led differentiated care models show promise but remain underexplored. We examined multi-stakeholder perceptions within the I'mPossible Fellowship, a peer-support DSD intervention addressing APHIV health, education, and livelihoods From May-December 2023, we enrolled three stakeholder groups: (1) intervention deliverers (APHIV "fellows" 18–27 yrs), (2) facilitators ("supervisors" of APHIV), and (3) recipients(APHIV "peers" in care, 8–26 yrs). We conducted interviews with 8 fellows (75% female, mean age 22.5) and 7 supervisors; and three focus group discussions with 18 peers (mean age 16.7). Audio-recorded data were transcribed, translated, and thematically analyzed using deductive coding and triangulation across groups to examine fellows' roles and program impact. Five key themes emerged, highlighting the layered influences of the I'mPossible Fellowship. First, mentorship provided informational and emotional support for peers; second, peer influence arising from peer-to-peer interactions contributed to a sense of trust and affirmation. Third, fellows' personal growth stimulated their motivation to fulfil their mentorship roles effectively. Fourth, stigma and poor preparedness for transition into adult care hindered educational, employment and independent living opportunities for APHIV. Fifth, sustainability, through continued mentoring support, was emphasized by supervisors and fellows as crucial for supporting APHIV in transitioning to independent

**Data availability statement:** This study involved qualitative interviews and focus group discussions with youth living with HIV and program stakeholders in India. While anonymized excerpts from transcripts have been included within the manuscript to illustrate key findings, we did not obtain ethical approval to share full transcripts publicly. Participants were informed during the consent process that transcripts would be stored securely and destroyed following the completion and dissemination of the study. The Johns Hopkins Bloomberg School of Public Health IRB and the YR Gaitonde Centre for AIDS Research and Education (YRG CARE) IRB approved the study with these conditions. Public sharing of full transcripts would violate these approved protocols and pose significant ethical risks, potentially compromising participant confidentiality and safety. The code book from this analysis has been included as supplementary information. This document lists all the codes that our data analysts used and their code definitions where they were applied. Researchers with specific questions or requests related to the raw data or data analysis may contact the corresponding author at ashet1@jhu.edu, or reach out to the IRBs at jhsph.irboffice@jhu.edu or ethics@yrgcare.org.

**Funding:** This work was supported by National Institutes of Health (K23DA057151 to LG), Rishi Children's Fund, Department of International Health, Johns Hopkins Bloomberg School of Public Health (AS). The funders had no role in study design, data collection and analysis, decision to publish, or preparation of the manuscript.

**Competing interests:** The authors have declared that no competing interests exist.

**Abbreviations:** APHIV, adolescents and young adults with perinatally acquired HIV; ART, antiretroviral treatment; CCI, childcare institution; DSD, differentiated service delivery; FGD, focus group discussion; HIV, human immunodeficiency virus; IDI, in-depth interview; SEM, socio-ecological model; YLHIV, youth living with HIV.

living. This study highlights the pivotal role of fellows and peer mentorships in addressing the multilevel factors that enhance outcomes for APHIV. By providing knowledge and empathy to their peers and serving as credible role models with lived experience of HIV, fellows within the I'mPossible fellowship exemplify a successful DSD model incorporating the three essential attributes of peer support: informational, emotional, and affirmative support. These findings underscore the importance of integrating peer-led interventions into HIV care and reframes youth as active agents of change, recognizing their capacity for meaningful societal contribution.

## Introduction

An estimated 3.1 million adolescents and young adults ages 15–24 years old were living with human immunodeficiency virus (HIV) globally in 2023 [1]. With the advancement and increased availability of antiretroviral treatment (ART) options, a more significant number of youth living with HIV (YLHIV) are surviving into adulthood [2,3]. This demographic confronts profound psychosocial and health-related challenges, such as the compounded stress of managing a stigmatizing, chronic illness, alongside experiences of parental loss, financial instability, and social discrimination, that significantly exacerbate their vulnerability [4–8]. Transitioning into adult healthcare systems frequently presents logistical and emotional hurdles that can further complicate long-term care engagement [7,9,10]. These adversities manifest in reduced adherence to ART and poor viral suppression rates, leading to higher rates of treatment failure, morbidity, and mortality compared to other age groups [11–14].

HIV care models have sought to address these multifaceted challenges among YLHIV using differentiated service delivery (DSD) models in HIV care [14]. These models adapt the provision of services-such as medication dispensing, counseling, and follow-up care-to better align with their unique lifestyles, preferences, and needs, offering a more person-centered approach to healthcare. Peer support has emerged as an effective strategy to tailor services to individual needs, as it leverages shared lived experiences to foster trust, reduce isolation, and provide both practical and emotional guidance [12,15–17]. This support can be leveraged through community-based DSD models, as opposed to facility-based ones, and is more advantageous in improving accessibility by improving barriers such as travel costs and rigid scheduling [18–20]. This positions YLHIV not merely as disadvantaged recipients of care but as individuals with agency capable of being empowered to contribute meaningfully as social assets within their communities.

Research from sub-Saharan Africa highlights the potential of peer support interventions to improve viral suppression and ART adherence among YLHIV. However, there are critical gaps in the existing literature [19,21]. First, while existing peer support models demonstrate progress in improving health outcomes among YLHIV, their broader psychosocial, educational, and livelihood needs extend beyond medical care and are often overlooked in research and programmatic implementation. Second, in many HIV peer support programs, a recurring challenge is the limited focus on "caring

for the carer," referring to the emotional and practical needs of the peer supporters who deliver care and guidance to their communities [22]. While these carers provide substantial support to YLHIV, programs often neglect strategies to sustain their well-being, leaving them vulnerable to emotional exhaustion. Third, studies on peer support are centered mainly around experiences in the Sub-Saharan Africa region, while the YLHIV population in India remains under-recognized. India has the largest population of YLHIV in Asia, with national estimates from 2023 indicating 163,000 adolescents and young adults aged 15–24 years are living with HIV [2,23,24]. Additionally, the residential status of many YLHIV is shaped by parental loss or incapacity, with a proportion residing in childcare institutions (CCIs) as a result [25]. CCIs are residential facilities that provide care, maintenance, and supervision for minor children. In the context of our study setting, CCIs refer to residential facilities caring for children who have lost one or both parents due to HIV. Most children living in the CCIs in our setting are born with HIV themselves. India has a large proportion of orphaned children due to HIV; in a six-year retrospective analysis of national-level data from six high-HIV-prevalence Indian states, from among 9,051 children with HIV, orphans constituted 55%, with 36% and 19% having lost one or both parents, respectively; 95% lived with parents or relatives, and 4% resided in child-care institutions [26]. In particular, a large proportion are adolescents and young adults with perinatally acquired HIV (APHIV), a distinct subgroup of youth with HIV diagnosed in early childhood.

Our study addresses these three gaps-support beyond health needs, focus on the intervention deliverer, geographic gap in India-by studying the I'mPossible fellowship program, a peer-led intervention for APHIV in India. Guided by the Social Ecological Model of health (SEM), the I'mPossible Fellowship recognizes that youth well-being is shaped by multiple, interacting levels of influence, from individual resilience, interpersonal peer support and organizational structures. In this study, we aimed to examine experiences of the first batch of fellows initiated within the I'mPossible fellowship, who provided peer support (interpersonal layer), and triangulate findings with qualitative data obtained from peers as beneficiaries of this support (individual layer) and from institutional supervisors (organizational layer) who function as informal mentors to fellows. Through this well-rounded qualitative exploration, this research aims to reveal the practical realities of implementing peer-led DSD models and-the potential of these models to inform improved HIV care in India and similar settings.

## Methods

### Study setting

This study was conducted in six districts across the south Indian states of Karnataka and Tamil Nadu (Bangalore, Belgaum, Krishnagiri, Gulbarga, Kolar, Vijaypura, and Raichur) between 3 May 2023–24 November 2023. Procedures were executed in collaboration with local partners, non-profit organizations, families of APHIV, and childcare institutions (CCIs) that offer residential support and care for APHIV in these regions.

### The I'mPossible fellowship and the social-ecological model of health

The I'mPossible fellowship intervention, launched in 2021 in the southern Indian states of Karnataka and Tamil Nadu, is a peer-support DSD model designed and adapted to support APHIV. Using a hybrid delivery model comprised of one-on-one sessions and group sessions conducted over a 48-week period, participant beneficiaries referred as "peers", include children, adolescents, and young adults aged 8–26 years living with HIV and engaged in HIV care. Peers are provided with tailored health, educational, and livelihood support and mentorship. These services are provided by "fellows" - trained youth living with HIV aged 18–27 years. Fellows are recruited based on their demonstrated motivation and leadership potential. They engage in a month-long in-house training at a partner institution based on sessions adapted from the International Center for AIDS Care and Treatment Programs curriculum [27]. Following the training, fellows are assigned approximately 25 peers residing either in CCIs or within the community in family-based care. Based on their location, each fellow serves as a peer mentor and works under the guidance of an experienced supervisor who provides oversight and support. Fellows are supported through stipends and encouraged to maintain their health and pursue higher education or

vocational training. This dual-focus approach ensures the program's sustainability by prioritizing fellows' well-being and professional development while enabling them to act as agents of change within their communities.

The I'mPossible fellowship intervention is grounded in the socio-ecological model (SEM) of health, which serves as a person- and context-centered intervention framework [28]. This framework addresses the individual while emphasizing the multiple spheres of influence (interpersonal, organizational, community, and public policy) on health and well-being [28]. The fellowship focuses on three levels - individual, interpersonal, and organizational - and integrates stakeholders within each domain (Fig 1). Peers represent the individual or personal level, reflecting their unique knowledge, attitudes, and skills in self-managing HIV. Fellows embody the interpersonal level, highlighting the interactive dynamics and support they provide to peers. Supervisors, who often fulfill the role of guardians for peers and as informal mentors to fellows, operate within organizational and supervisory structures such as HIV clinical care centers, religious communities, educational institutions, and non-profit organizations representing the organizational levels of influence.

### Study participants

This study presents qualitative accounts from three key stakeholder groups involved in the I'mPossible fellowship - fellows, peers, and supervisors, each representing a distinct perspective within the socio-ecological model (Fig 1). In this study, the term "YLHIV"' is used to refer to all youth living with HIV, whereas "APHIV" denotes the subset of YLHIV who

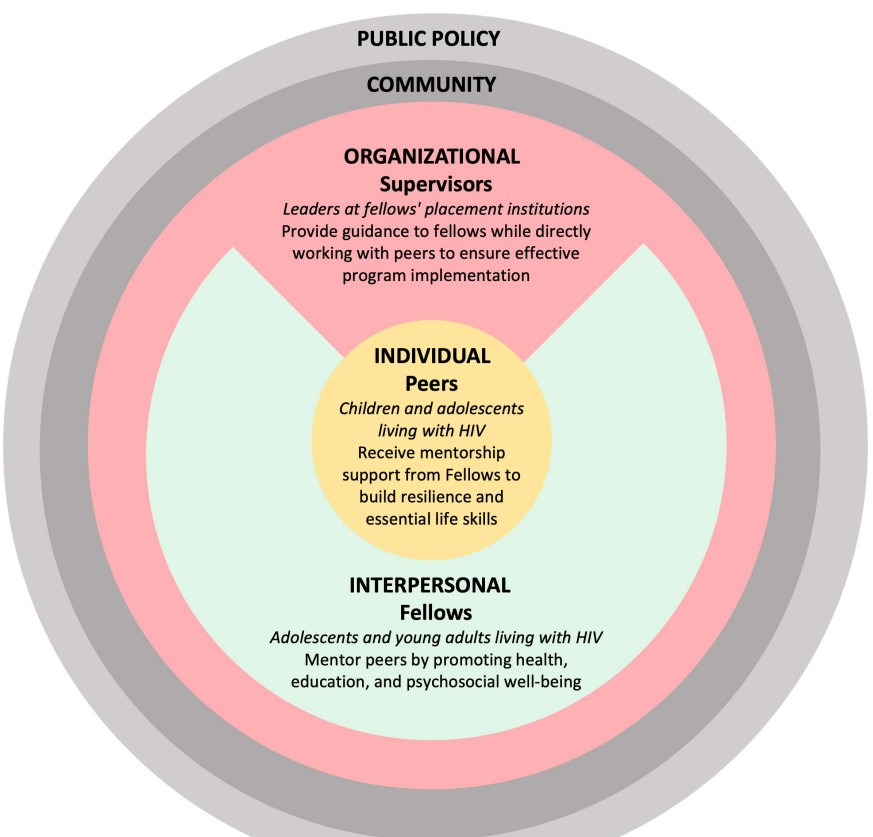

**Fig 1. Study participants' perspectives presented through the lens of the socio-ecological model framework.**

acquired HIV perinatally, defined as those diagnosed with HIV prior to the age of 10 years. All fellows and peer participants in this study are APHIV by this definition.

Eligible fellows were those from the first cohort of the I'mPossible fellowship, and peers were APHIV aged ≤18 years who maintained regular contact with at least one fellow from the I'mPossible intervention for a minimum of three months. Supervisors were purposively selected from faculty or staff members at CCIs, non-governmental organization managers, and healthcare workers who directly supervised the fellow for at least three months. Among our 18 study participants, 15 (83.3%) resided in CCIs, while the remaining 3 (16.7%) lived in community-based households. Fellows were compensated for their participation, but not supervisors and peers.

## Ethics approval and consent to participate

The study received approval from the Institutional Review Boards of the YR Gaitonde Centre for AIDS Research and Education in Chennai, Tamil Nadu (#YRG375) and the Johns Hopkins Bloomberg School of Public Health in the United States (#IRB00023077). Informed consent for this study was obtained orally due to the combination of in-person and telephonic data collection, which made obtaining written consent logistically challenging, as virtual participants often lacked access to printers or scanners to provide signed consent forms. Oral consent, assent, and parental permission were obtained through researcher-signed documentation, which confirmed that informed consent was provided and that participants voluntarily agreed to participate in the study. Adult participants provided oral informed consent for their participation. For participants <18 years, oral parental permission was obtained from their parents or legal guardians, and oral assent from the minor participants themselves. All consent, assent, and parental permission forms were available in English and the local language, Kannada. Institutional Review Boards approved the use of oral consent, assent, and parental permission ensuring that participants received a clear explanation of the study's purpose, procedures, risks, and benefits. Additional information regarding the ethical, cultural, and scientific considerations specific to inclusivity in global research is included in the Supporting Information.

## Data collection

To support the multi-perspective qualitative research on the impact of the I'mPossible intervention, we first conducted individual in-depth interviews (IDIs) with 8 fellows (75% female; mean age: 22.5 years; age range 18–24) and 7 supervisors to (1) explore fellows' motivations, leadership development, and overall experiences within the intervention, and (2) examine the roles and contributions of fellows, their perception of the influence of the intervention on peers, specifically assessing observable benefits or changes in the fellows and peers. We employed IDI as the exploratory tool among fellows and supervisors in order to obtain individual-level attitudes, knowledge, and perceptions. Interviews were conducted in English by a trained interviewer (SS) either over the phone or at a location preferred by the participants and lasted approximately 1–2 hours per participant. The sample size of 15 in-depth interviews, 8 fellows and 7 supervisors, reflects the total number of eligible participants available in the program during the study period as we restricted recruitment to the first batch of fellows within the I'mPossible fellowship intervention from 2021. With the exception of one additional fellow who was unavailable for interviewing, all individuals in these roles were included. As such, the sample represents near-complete coverage of the participant population, and thematic saturation was achieved within both groups. All fellows and supervisors were proficient in English.

Next, we conducted three focus group discussions (FGDs) with 18 peers (66.7% male; mean age: 16.7 years; age range: 12–18 years) to characterize their interactions with fellows, the program's impact on their quality of life, and the challenges of living with HIV. FGDs were selected as the primary method for engaging peers due to their developmental appropriateness and feasibility. While IDIs are often used to explore individual-level perceptions, FGDs were chosen to better capture the collective experience of being a peer in this program, as well as to encourage open discussion and peer interaction. The majority of peers (n = 15, 83.3%) resided in CCIs, while the remaining (n = 3, 16.7%) lived in

community-based households. Educational attainment varied: six participants were pursuing 12th grade, six were in 11th grade, two in 10th grade, two in 8th grade, one in 6th grade, and one was enrolled in an Industrial Training Institute (ITI) program. FGDs were facilitated by trained India-based researchers (MBR, EN) in Kannada and English. Interview guides for IDIs and FGDs were semi-structured and iteratively developed with input from program staff and youth investigators from the community (BS, SR). All sessions were audio-recorded with participants' written informed consent. The research team (SR, SS, EN) transcribed the audio recordings, and translation services were facilitated by a local translator based in Karnataka.

## Data analysis

The research team coded and analyzed the IDI and FGD transcripts using Dedoose 9.0 software. Two complementary methods of analysis were employed to assess the multiple-perspective qualitative data. First, conducting a thematic analysis, following the methodology outlined by Braun & Clarke (2006) [29]. Independent coders (SS, AAS, AK) carried out the analysis in six iterative steps: they familiarized themselves with the data, generated initial codes, searched for themes, reviewed themes, defined and named themes, and produced the final report for each data collection method. IDIs with fellows and supervisors were analyzed to explore individual-level experiences and reflections, while FGDs with peers were analyzed separately, with attention to group-level dynamics, shared experiences, and interactions between participants. Second, they applied the analytical steps for triangulating qualitative multiple-perspective interviews, as described by Vogl, Schmidt, and Zartler (2019) [30]. This approach further examined the coded data generated through the thematic analysis across different participant groups (fellows, peers, and supervisors) utilizing the SEM model to identify patterns, similarities, and contrasts among their perspectives [28]. To ensure that participant experiences were accurately captured, the researchers conducted member checking, also known as participant validation, for all data collection activities. Additionally, researchers engaged in reflexivity throughout the research process through qualitative memo writing and regular debriefing sessions to discuss the potential influence of their positionality on data collection and analysis.

## Results

Fellows, supervisors, and peers provided converging yet distinct perspectives on the core themes of the I'mPossible fellowship intervention. While all three groups recognized the importance of mentorship and emotional support within the fellowship, their perspectives highlighted specific insights regarding the intervention and generated recommendations on how the fellowship can better support the APHIV community. Their collective experiences revealed the fellowship's multi-layered effects through five core themes: (1) mentorship, (2) peer influence, (3) personal growth, (4) transition challenges, and (5) sustainability (Tables 1 and 2).

*Mentorship:* Mentorship, built foundationally on trust, was central to the fellowship's impact. Fellows stepped into dual roles as both teachers and learners, providing emotional and practical guidance while also bidirectionally benefiting from mentorship by supervisors and program staff. Peers valued this dynamic and regarded fellows as mentors, frequently seeking their guidance not only for medical and HIV-related issues, but also for educational and emotional support. More than just sources of information, fellows created safe spaces where peers felt genuinely heard and understood. In these spaces, difficult conversations could unfold without fear of stigma, allowing peers to openly share their personal struggles and aspirations.

*"They [fellows] were thinking of us and our health, remembering us... they care about us a lot."* - peer, female

Supervisors, acknowledging the depth of these relationships, emphasized the importance of providing consistent mentorship and capacity-building support. They highlight how, once supported, fellows become role models themselves, leading by example and using their own experiences to guide others. As one supervisor described, by openly sharing their own experience with medications, a fellow eased peers' fears about ART-related side effects.

PLOS Global Public Health

**Table 1. Main themes and subthemes from fellows' perspectives with exemplary quotes.**

| Subtheme | Fellows' Quotes |
| --- | --- |
| **Mentorship** | |
| Peer mentorship | *"Every girl needs somebody in their life, to really have support… any girl needs at least a good friend, or a good mother, sister, somebody who she can really trust."* - fellow, female |
| Mentorship by mentors and supervisors | *"Actually, being with my mentor helped me a lot… He showed me, then he asked me how to do it, and he corrected me. I started doing it, and then he started giving me all the responsibilities of a fellow, where I can be mentored, not like that."* - fellow, male |
| **Peer influence** | |
| Emotional support and belonging | *"They [peers] said that they are also HIV positive and that they want to speak in front of other HIV-positive people. They said, I also want to become like you, akka [sister]. When you were explaining this problem, we felt more comfortable and relaxed, that made me feel happy."* - fellow, female |
| Empathy fatigue | *"… I remembered my background, how they treated us, how my family treated me. I cannot give any suggestions to her, but still, I kept myself strong. And I told her, 'It's okay. You grow up a little more.'"* - fellow, female |
| **Personal growth** | |
| Motivation | *"I can be a source of motivation for all the HIV clients."* - fellow, female |
| Resilience | *"Before, I was like thinking so much about things, but now I don't have that fear in me. The work that is said to me is also done on time, and I visit different centers. Before, I was also scared to speak to others, but now I don't have that fear in me, and I also travel alone to those places where I am asked to go."* - fellow, female |
| **Transition challenges** | |
| Stigma and discrimination | *"Can't say for others that I'm having HIV... they will be themselves and they won't mingle with others. And even parents were telling not to tell…"* - fellow, female |
| Educational barriers | *"Till today, from the time I started working, my family doesn't give me even a single penny. I work, I save that money, and use it for my purposes when I go home."* - fellow, female |
| Mental health and emotional well-being | *"I was also abused at a very young age, so I know how the pain is… There were so many girls where I was working, and so many girls from here; at least there'll be one person who has gone through that stage, maybe they were abused in a different way, or it can be sexually or mentally. Those scars are very hard to share. They have many difficulties and cannot focus on anything."* - fellow, female |
| Lack of practical skills | *"Hardest part was that I was not able to complete the reports on time."* - fellow, female |
| Role ambiguity | *"No communication... none of her [peers] know that she is a fellow. They think that she is a staff that came for new admission. They think that she is a teacher or warden."* - fellow, male |
| **Sustainability** | |
| Career guidance | *"If you want to teach them what life outside is like, somebody who has experienced life outside has to come and explain their experience."* - fellow, male |
| Extended support networks | *"So, for the next five years, I am openly stating that I will support students-not through the fellows, but personally."* - fellow, male |

*"Say, like, it was practical examples for the children when the fellow was saying something or doing something, especially with their ART drugs. She herself was giving her own examples and telling them, 'See, these things [side effects] should happen, and I have been taking these tablets [ART] for so many years."* - supervisor 1

**Peer influence:** Shared experiences between fellows and peers fostered a sense of belonging and emotional support. Peers described that the understanding derived from similar life journeys created an environment where they felt seen, heard, and inspired, and accepted them as positive role models who mirrored their challenges.

*"We feel less stressed when we speak to the HIV positive person and share our problems with them and we also feel more comfortable with them."* - peer, male

As a result of this connection, peers felt more receptive to encouragement and advice on healthy behaviors, such as adhering to their medication and staying physically active. Supervisors reinforced this by noting that fellows' lived

**Table 2. Themes and exemplary quotes from supervisors and peers, organized by sample.**

| Theme | Finding | Supervisor Quote | Peer Quote |
|---|---|---|---|
| **Mentorship** | Mentorship fosters informational, emotional and academic support. | *"Say, like, it was practical examples for the children when the fellow was saying something or doing something, especially with their ART drugs. So she herself was giving her own examples and telling them."* - supervisor 1 | *"She [fellow] was helping us in our studies."* - peer, male |
| **Peer influence** | Shared lived experiences build trust and engagement. | *"…these people [fellows] are always available, and at the same time, they are ready to understand their own person [peer] who is suffering from the same difficulties."* -supervisor 3 | *"With [HIV] positive people [such as fellows], we can speak much more freely."* -peer, female |
| **Personal growth** | Fellows develop leadership and resilience through carrying out their responsibilities. | *"It was a real learning for her, like, see, even our fellows, like, they were just like the children previously in the institutions, and now they themselves are monitoring this."* - supervisor 1 | *"[If we were to become fellows] We will get confidence by teaching others and help others by telling them to take tablets."* - peer, male |
| **Transition challenges** | Stigma and educational barriers create uncertainty about the future, which may impede outcomes. | *Stigma and negative ideas are there, and nobody is educating them. Nobody is bringing awareness. Awareness is there for the ones who are positive, but not for the family members…* - supervisor 2 | *"First we were studying in another school and once they came to know about our status, then we were put in another school."* - peer, male |
| **Sustainability** | Strengthened community connections support long-term success. | *"I feel we should somehow gather them once in a while or in some way we have to just give them…, mentoring and help helping them to stay connected, all that, let them know they have a community."* - supervisor 5 | *"Now we are in [CCI]. Here we get support but after we go out from [CCI] where will we get support from?"* -peer, male |

experiences with HIV enabled them to form more profound and meaningful relationships with their peers than staff members. This was explained by one supervisor who emphasized that only someone else living with HIV ("another positive") can truly understand the experiences of a person living with HIV.

> *"Another positive can be understood by one more positive only better. Whereas for other staff, there may be some stigma or difficulty in understanding unless they have a good counseling course or experience over the years."* - supervisor 3

While these close connections provided essential support, they also burdened fellows emotionally. The empathy that made them effective mentors also made them vulnerable to emotional exhaustion, as fellows often found themselves reliving their hardships through the experiences of their peers.

> *"She's [peer] the only daughter, and her mother died at a very young age. I don't have any solution because my family has the same problem. I don't have property. I too don't have parents. So, I just started crying with her."* - fellow, female

**Personal growth:** Fellows indicated that personal growth is a dynamic process. Those who initially struggled with self-doubt, found strength in their relationship with others, cultivating resilience, specifically relational resilience, by leaning on the support of their peer network. One fellow described how participation in the program helped them overcome their fears and gain more self-assuredness.

> *"Before, I was like thinking so much about things, but now I don't have that fear in me. The work that is said to me is also done on time, and I visit different centers. Before, I was also scared to speak to others, but now I don't have that fear in me, and I also travel alone to those places where I am asked to go."* - fellow, female

This growth was supported by a desire to uplift others, fellows began to recognize themselves as role models. As one fellow reflects, they came to understand that their journey could serve as a powerful example for peers navigating a similar path.

*"I can be a source of motivation for all the HIV clients [peers]."* - fellow, female

Supervisors also witnessed this transformation firsthand, observing fellows evolve from uncertain participants into confident leaders. They not only observed fellows acquiring practical skills such as time management, effective communication but also a deeper understanding of themselves and their community. One supervisor recounted the growth of a fellow, who was previously in a childcare institution, expressing both surprise and pride in their transformation.

*"Of course, the thing initially, I used to think, how would [fellow] go out and make a life by herself? You know, like she used to be always very much dependent… But when we gave her an opportunity, like, okay, you can go out and work, like there is this type of work and that- it was to my wonder that she managed it so well. Like, initially, she used to keep calling and say, 'This is difficult… all that, but still, she managed."* - supervisor 5

Peers, too, perceived this potential for growth in the fellow role, seeing it as an opportunity for empowerment, education, and self-improvement. Those who once received support were eager to step into guiding roles themselves, reflecting an aspirational cycle of mentorship.

*"[If we were to become fellows] We will get confidence by teaching others and help others by telling them to take tablets [ART]."* - peer, male

**Transition challenges:** Fellows often grappled with financial instability and limited access to educational opportunities, which led them to highlight the need for career guidance and exposure to life beyond the program. Like the fellows, peers expressed uncertainty about their future, particularly transitioning out of the structured environments of CCIs and the fellowship.

*"After finishing 12th, we are not sure if we will stay in [CCI] or not, and if they suddenly send us out and we cannot find a job, we will be alone. We are not aware of what support we will be getting."* - peer, male

This uncertainty is exacerbated by the fact that, as Fellows observed, their peers faced numerous challenges that could jeopardize a successful transition into adulthood.

*"In this area, child marriage they'll do. Like seven years they'll do, then girls they won't go school. Boys also, in second standard, they'll say, 'I don't want to go to school,' then their family members won't allow them to go, and they'll just sit there. Education problems are there, marriage problems, and financial problems."* - fellow, female

Supervisors also recognized these concerns for successful independent living, identifying transition planning gaps, particularly job placement and long-term mentorship. They also brought to attention the need for more community facing initiatives to address stigma and educate family members of those living with HIV.

*"Stigma and negative ideas are there, and nobody is educating them. Nobody is bringing awareness. Awareness is there for the ones who are positive, but not for the family members. So, what I see is that sometimes, even after knowing, attempts are made to get them married soon. Because of the stigma, they keep them away or push them away or see to it that they go out of the family. The fellowship has to reach out to the positive family members and bring awareness; that is the important thing, what I feel."* - supervisor 3

Additionally, although fellows did develop practical skills throughout the fellowship, they still, at times, felt underprepared to fulfill their responsibilities, especially in the early stages. They described initial challenges with facilitating peer support, leading sessions, and completing administrative tasks such as writing reports, often struggling to find confidence

in these new roles. Supervisors also observed these difficulties and recommended setting clearer expectations and providing more structured preparation for fellows' roles during the training process.

*"I have already told them that they should know what program they will be doing; some kind of timetable or agenda should be there. If they come to the house, what will they be doing? What activities will they be conducting for the children on a weekly or monthly basis? Something concrete was not done, so I was asking them to keep that ready for her to follow up on regularly."* - supervisor 2

**Sustainability:** Like the fellows, peers expressed uncertainty about their future, particularly transitioning out of structured environments of CCIs and the fellowship.

*"Now we are in [CCI]. Here we get support but after we go out from [CCI] where will we get support from?"* - peer, male

Supervisors further identified transition planning gaps, particularly career placement and long-term mentorship. They suggested that establishing a system for post-fellowship support would help fellows and peers successfully transition to life after the fellowship and maintain the connections they built during the program.

*"I feel we should somehow gather them once in a while or in some way we have to just give them…, mentoring and help helping them to stay connected, all that, let them know they have a community."* - supervisor 5

Continuing this cycle of support, peers generally wanted to stay connected with the program and become future fellows. However, some had reservations about taking on a fellowship role due to concerns about their capabilities or the potential stigma associated with disclosing their HIV status.

*"Fellows talk to everyone and share and I don't freely share with others and that's why I can't work like them… Sometimes I want to but on the other side I feel nervous."* - peer, female

Additionally, fellows expressed a desire to remain connected with their peers even after their formal responsibilities ended. Across all program stakeholders, there was a shared emphasis on sustaining this community beyond the program's official timeframe.

*"So, for the next five years, I am openly stating that I will support students- not through the fellowship, but personally."* - fellow, male

Across all participant groups, there was a common belief that the I'mPossible Fellowship fostered mentorship, emotional support, and youth empowerment. Their perspectives provided insight into the key differences in how these themes were experienced and enacted. While these perspectives were distinct, they were not conflicting, rather, they were complementary insights shaped by participants' roles within the fellowship (Tables 1 and 2).

## Discussion

The key findings from this qualitative study highlight how the I'mPossible Fellowship fosters personal growth, validation, and self-confidence among APHIV, benefiting both the peers (intervention recipients) and the fellows (intervention providers). (Fig 2). The fellowship, reinforced by peer support, ensures that services are accessible and responsive to the unique needs of APHIV. Our findings suggest that this model of peer support can lead to improved health behaviors, reduced stigma, better coping mechanisms, and a smoother transition into adulthood, resulting in sustained community

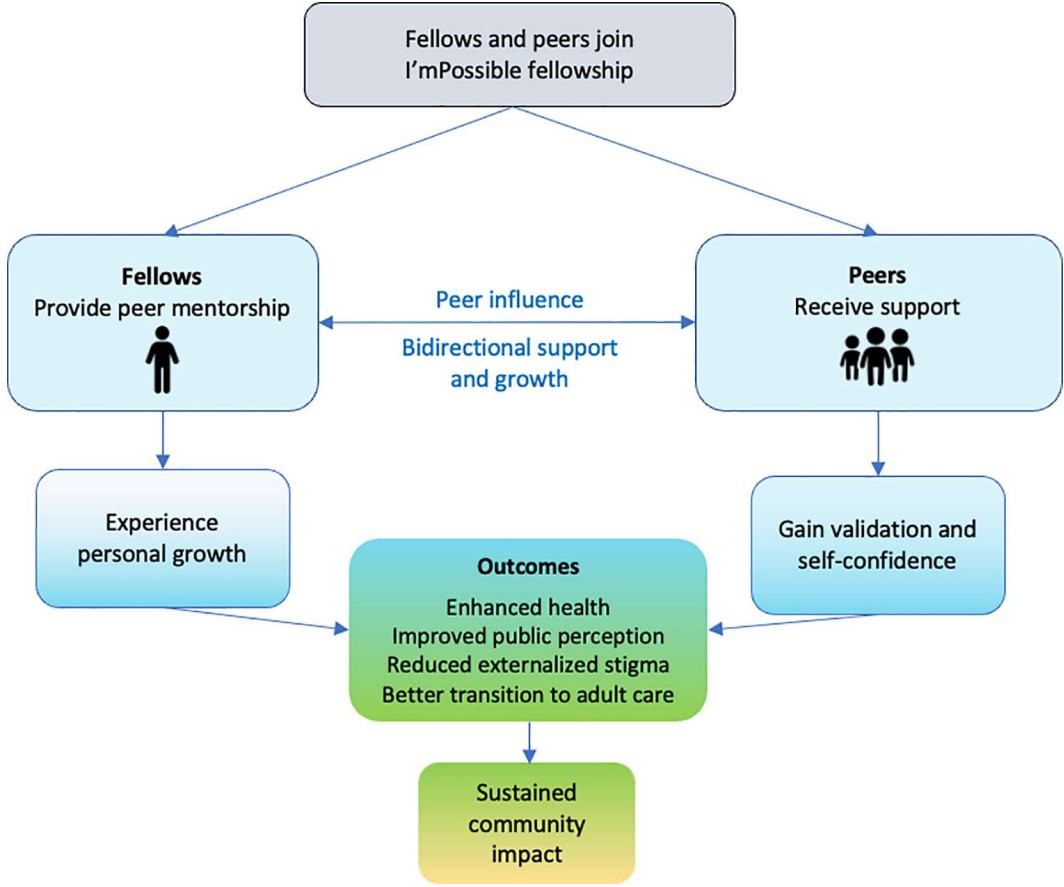

**Fig 2. The journey of fellows and peers throughout the I'mPossible fellowship: insights from the study findings.**

impact (Fig 2). This holistic, multidimensional approach underscores the value of integrating peer mentorship into DSD frameworks to enhance long-term health outcomes and social well-being through two unique features: (1) service delivery that is more inclusive of other non-health-related domains and (2) a greater focus on the well-being of the individuals providing support.

The I'mPossible fellowship, while aligned with known DSD frameworks, distinguishes itself through integrating education, vocational training, and community-based support into the peer mentorship framework, an evolution into what could be considered an advanced DSD approach [31]. Earlier DSD models were designed to decentralize HIV services, promote task shifting, and enhance accessibility to medical treatment. Our approach, however, is expanding the definition of DSD to be more multilevel, person-centered, and community-driven, recognizing that sustained engagement in care requires interventions beyond just medical treatment [31,32]. This next-generation DSD model acknowledges that social determinants of health, including poverty, stigma, mental health, and educational disparities, directly impact treatment adherence and overall well-being [32,33]. As global HIV response efforts evolve, implementing holistic, youth-centered DSD models, like the I'mPossible fellowship, will be critical to ensuring that every YLHIV has the tools, networks, and opportunities to thrive.

A key feature of the I'mPossible fellowship is its emphasis on the well-being and sustainability of the peer mentors themselves. Unlike traditional models focusing solely on mentee outcomes, the fellowship recognizes that effective peer

mentorship requires "caring for the carer" strategies [22]. Fellows, many of whom have faced their own challenges with stigma, mental health, and treatment adherence, regardless of their leadership abilities, require sustained emotional support from supervisors and access to professional development opportunities to prevent emotional exhaustion. This approach strengthens the overall effectiveness of peer-led interventions and ensures that mentors can model resilience and self-efficacy for their peers.

Mentorship is the foundation of the fellowship and is driven by fellows' sharing of lived experiences to provide emotional support and practical guidance to their peers. The bond that emerges, as a result, fosters a sense of psychological safety, which enables peers to discuss their challenges without fear of judgment and fellows to feel empowered to share their stories. This relational depth of mentorship emerged organically, reinforcing the essential components of peer support: informational, emotional, and affirmational support [34]. Fellows provide informational support by educating peers on HIV management strategies, emotional support by offering empathy and encouragement, and affirmational support by validating traumatic and emotional experiences.

However, these bidirectional benefits, coupled with complex, systemic barriers, can hinder the sustainability and scalability of peer mentorship in HIV care. Stigma and discrimination, for example, deterred many APHIVs from accessing care. While fellows often acted as intermediaries between youth and healthcare services, fears of being discriminated against persisted in schools, workplaces, and residential living. Many APHIV anticipated facing limited education and employment opportunities because of prevailing community attitudes towards HIV. Evidence from existing literature suggests that this vulnerability can compel APHIV to prioritize immediate needs, such as financial stability over adherence and retention to care, threatening their long-term health management [33]. These transition challenges can be addressed by integrating educational and vocational support to ensure peer mentorship translates into long-term sustainable health and social outcomes.

These findings align with global peer mentorship models that have successfully leveraged the power of trust and relatability of peers to improve health outcomes among YLHIV, although there are limited examples. Zimbabwe's Zvandiri program has effectively integrated community adolescent treatment supporters (CATS) into healthcare settings, providing psychosocial support, health education, and care linkage services [19,22]. Similarly, Zambia's Project YES! saw measurable improvements in viral suppression and mental health outcomes after placing peer mentors in HIV clinics [21,35]. These programs reinforce that peer-led approaches within DSD models support positive treatment outcomes, adherence, and well-being for YLHIV in low-and middle-income settings.

A key strength of this research design involves the inclusion of three levels of stakeholders, one from each of the first three layers of the SEM. A comparative analysis across these participant groups (fellows, peers, and supervisors) revealed layered experiences that demonstrate how the Fellowship operated at multiple levels of the SEM - individual, interpersonal, and organizational (Fig 1). A second methodological strength is the use of both IDIs and FGDs within the same study population, which enabled a richer, more nuanced understanding by capturing both individual experiences and collective perspectives. The combination of the two complementary qualitative methods, known as intra-paradigm research, revealed unique variations in power, participation, and program engagement across stakeholders, which would have otherwise been overlooked if this study had relied on a single method or participant perspective [36]. The IDIs with fellows and supervisors elicited individualized reflections on the program implementation and individual journeys of growth, while FGDs with peers revealed collective experiences of participation. Furthermore, these findings can be used to offer more specific, and actionable improvements to the I'mPossible Fellowship's design, training, and post-program support. Our study implemented an evaluation strategy that matched the complexity of social and behavioral interventions by embracing a multi-level, eco-system-based approach, moving beyond a single stakeholder or methodological perspective.

While our study offers valuable insights into youth-led interventions for HIV care, we acknowledge some limitations. Our research focused on the first batch of fellows within the initial 18 months of implementing the DSD model. As this was an early phase with

small numbers conducted when the model was still being refined, the experiences captured reflect preliminary lessons. This resulted in a sample size of only eight fellows who were interviewed, along with seven supervisors, although thematic saturation was achieved within both groups. The study is highly contextual, and findings may vary in different healthcare infrastructures or stages of DSD model implementation. Nevertheless, our study offers essential strengths, including its youth-centered participatory approach and the rich qualitative data drawn from the direct experiences of both mentors and mentees. These insights provide valuable guidance for refining peer-led mentorship models and tailoring DSD approaches to meet the evolving needs of YLHIV better.

In conclusion, the I'mPossible fellowship represents a significant advancement of community-based DSD models through the operationalization of structured peer mentorship. By integrating informational, emotional, and affirmational support with critical psychosocial, educational, and social dimensions, the program demonstrates how DSD can be expanded to best address the needs of YLHIV. Future research should focus on scalability, sustainability, and long-term impact, particularly in addressing economic barriers and stigma-related challenges that hinder optimal youth transition into adult care and successful integration into general society.

## Supporting information

**S1 Text. Qualitative codebooks.**
(DOCX)

**S1 Checklist. Inclusivity in global research.**
(DOCX)

## Acknowledgments

The authors acknowledge the RISHI Foundation and Sneha Charitable Trust, as well as the participating childcare institutions and participants, for their contributions to this research.

## Author contributions

**Conceptualization:** Siddha Sannigrahi, Michael Babu Raj, Babu Seenappa, Suhas Reddy, Satish Kumar SK, Anita Shet.

**Data curation:** Siddha Sannigrahi, Michael Babu Raj, Babu Seenappa, Ashley A. Sharma, Suhas Reddy, Esha Nobbay, Satish Kumar SK, Anita Shet.

**Formal analysis:** Siddha Sannigrahi, Michael Babu Raj, Ashley A. Sharma, Suhas Reddy, Esha Nobbay, Aastha Kant, Lakshmi Ganapathi, Anita Shet.

**Funding acquisition:** Anita Shet.

**Investigation:** Siddha Sannigrahi, Michael Babu Raj, Babu Seenappa, Ashley A. Sharma, Suhas Reddy, Lakshmi Ganapathi, Anita Shet.

**Methodology:** Siddha Sannigrahi, Michael Babu Raj, Babu Seenappa, Ashley A. Sharma, Aastha Kant, Baldeep K. Dhaliwal, Lakshmi Ganapathi, Anita Shet.

**Project administration:** Siddha Sannigrahi, Michael Babu Raj, Babu Seenappa, Anita Shet.

**Supervision:** Siddha Sannigrahi, Lakshmi Ganapathi, Anita Shet.

**Visualization:** Siddha Sannigrahi.

**Writing – original draft:** Siddha Sannigrahi, Anita Shet.

**Writing – review & editing:** Siddha Sannigrahi, Michael Babu Raj, Ashley A. Sharma, Satish Kumar SK, Baldeep K. Dhaliwal, Lakshmi Ganapathi, Anita Shet.

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
